# A Vector Bernstein Inequality for Self-Normalized Martingales

**Ingvar Ziemann**                                                          *ingvarz@seas.upenn.edu*
*University of Pennsylvania*

Reviewed on OpenReview: *https://openreview.net/forum?id=4ZJjr9YbBw*

## Abstract

We prove a Bernstein inequality for vector-valued self-normalized martingales. We first give an alternative perspective of the corresponding sub-Gaussian bound due to Abbasi-Yadkori et al. (2011) via a PAC-Bayesian argument with Gaussian priors. By instantiating this argument to priors drawn uniformly over well-chosen ellipsoids, we obtain a Bernstein bound.

## 1 Tail Bounds for Self-Normalized Martingales

Deviation inequalities for self-normalized martingales play a key role in obtaining guarantees for linear regression in interactive and sequential decision-making tasks, such as learning an autoregression or regret minimization in linear bandits. The most prevalent version of such an inequality currently in use is due to Abbasi-Yadkori et al. (2011) and rests on the method of pseudo-maximization popularized by Peña et al. (2009), dating back to Robbins & Siegmund (1970). Comparing their result to the central limit theorem, their bound is nearly optimal but depends on the sub-Gaussian variance proxy instead of the actual variance. In this note, we overcome this issue by casting the pseudo-maximization technique through the lens of the PAC-Bayesian inequality. The present approach simplifies classical pseudo-maximization by relegating the complexity of evaluating exponential integrals against the smoothing distribution to the computation of a generic Kullback-Liebler divergence term. This allows us to generalize the argument of Abbasi-Yadkori et al. (2011) but without access to globally defined moment generating function bounds.

Let us now proceed by describing the setting of our result and that of Abbasi-Yadkori et al. (2011). Fix a filtration $\mathcal{F}_{0:\infty}$ and two square-integrable processes: $X_{1:\infty}$, taking values in $\mathbb{R}^d$, and $W_{1:\infty}$, taking values in $\mathbb{R}$. For each $T \in \mathbb{N}$, let $X_{1:T}$ be adapted to $\mathcal{F}_{0:T-1}$ and $W_{1:T}$ to $\mathcal{F}_{1:T}$ with $\mathbf{E}[W_k|\mathcal{F}_{k-1}] = 0$ for every $k \in \mathbb{N}$. For $t \in \mathbb{N}$, we define:

$$S_t \triangleq \sum_{k=1}^{t} W_k X_k \quad \text{and} \quad V_t \triangleq \sum_{k=1}^{t} X_k X_k^{\mathsf{T}} \tag{1}$$

and are interested in bounds on the random walk $S_T$ in the random Mahalanobis norm $\|S_T\|_{(V_T+\Gamma)^{-1}}^2 = S_T^{\mathsf{T}}(V_T+\Gamma)^{-1}S_T$ for some fixed positive semidefinite matrix $\Gamma \succeq 0$. This is called a *self-normalized martingale*. Under the assumption that $W_k$ is $\mathcal{F}_{k-1}$-conditionally $\sigma_{\mathrm{subG}}^2$-sub-Gaussian for each $k \in \mathbb{N}$,[1] Abbasi-Yadkori et al. (2011) show that for every stopping time $\tau \ (\in \mathcal{F}_{0:\infty})$ and with probability $1 - \delta$:

$$\|S_\tau\|_{(V_\tau+\Gamma)^{-1}}^2 \le \sigma_{\mathrm{subG}}^2 \times \left[\log\left(\frac{\det(V_\tau + \Gamma)}{\det(\Gamma)}\right) + 2\log(1/\delta)\right]. \tag{2}$$

Their approach rests on an elegant application of the pseudo-maximization technique developed by Peña et al. (2009), dating back to Robbins & Siegmund (1970). While elegant, the result of Abbasi-Yadkori et al. (2011) has one shortcoming as compared to classical asymptotics: the linear dependence on the conditional variance proxy $\sigma_{\mathrm{subG}}^2$ as opposed to the conditional variance, $\sigma_{\mathrm{var}}^2 \triangleq \sup\{\mathbf{E}[W_k^2|\mathcal{F}_{k-1}]| \text{ a.s.}, k \in T\}$. The

---

[1] $\mathbf{E}\left[\exp\left(\lambda W_k\right) \big| \mathcal{F}_{k-1}\right] \le \exp\left(\frac{\lambda^2 \sigma_{\mathrm{subG}}^2}{2}\right), \ \forall k \in \mathbb{N}.$

traditional fix to this in the literature on concentration inequalities is to invoke a Bernstein-type bound on the moment generating function (MGF) of the $W_k$ instead of a Hoeffding type of bound (see e.g. Freedman, 1975; de la Pena, 1999). Unfortunately, directly combining a Bernstein MGF bound with the pseudo-maximization technique does not lead to analytically tractable upper bounds as it requires the evaluation of an exponential integral over a bounded domain (an ellipsoid). Moreover, previous attempts at establishing Bernstein bounds (via methods orthogonal to pseudo-maximization) for vector-valued self-normalized martingales suffer from extraneous logarithmic dependencies and have significantly looser constants (see e.g. Zhao et al., 2023).

In this note, we provide an alternative perspective on the proof of (2), where, instead of computing the exponential integral directly, we invoke the variational characterization of Kullback-Liebler divergence via the PAC-Bayesian lemma (Shawe-Taylor & Williamson, 1997; McAllester, 1998) to relegate this difficulty to the calculation of said divergence. Beyond Gaussian priors, this turns out to be significantly simpler than the evaluation of an exponential integral. It allows us to modularize the proof strategy of Abbasi-Yadkori et al. (2011) and replace their Gaussian prior with uniform ellipsoidal priors that are suitable in combination with exponential inequalities that only hold for a restricted domain (contrast this with Hoeffding MGF bounds being valid for all $\lambda$).

The rest of this note is organized as follows. We state our main result immediately below and then proceed to discuss its consequences. Its proof is given in Section 3.2. Preliminaries relating to our application of the PAC-Bayesian lemma are given in Section 3 where we also provide a proof of (2) as a warm-up. Auxiliary lemmata, including the PAC-Bayesian lemma, are proven in Section 4.

## 2 The Result

To apply a Bernstein MGF bound, we will require some additional boundedness assumptions. Namely, we posit that:

$$|W_k| \leq B_W \quad \text{and} \quad X_k X_k^{\mathsf{T}} \preceq B_X^2, \qquad \forall k \in \mathbb{N} \tag{3}$$

for a positive scalar $B_W$ and a positive definite matrix $B_X$. Our main result can now be stated:

**Theorem 1.** *Fix $\delta, \varepsilon, \nu \in (0,1)$, a stopping time $\tau$ with respect to $\mathcal{F}_{0:\infty}$, a positive semidefinite matrix $\Gamma \succeq 0$, a positive definite matrix $V \succ 0$ and assume that* (3) *holds. Define*

$$\alpha \triangleq \left( \frac{\sqrt{e}(1+\nu)\|S_\tau\|_{(V_\tau+\Gamma)^{-1}V(V_\tau+\Gamma)^{-1}}}{\nu\sqrt{d+2}} - 1 \right) \vee 0.$$

*Then as long as $V_\tau + \Gamma \succeq e(1+\nu)^2 V \succeq (1+\nu)^2 \varepsilon^{-1}(d+2)B_W^2 B_X^2$ we have that with probability at least $1-\delta$:*

$$\|S_\tau\|_{(V_\tau+\Gamma)^{-1}}^2 \leq \left( \frac{(1+\alpha)^2}{1+2\alpha} \times \frac{1}{1-\varepsilon} \right) \times \sigma_{\text{var}}^2 \times \left[ \log\left( \frac{\det(V_\tau+\Gamma)}{\det(V)} \right) + 2\log(1/\delta) \right]. \tag{4}$$

**Remarks on Theorem 1:**

1. The bound gives a refined confidence ellipsoid for least squares estimation with martingale difference noise in the model $Y_k = \langle \theta_\star, X_k \rangle + W_k$ up to a stopping time $\tau$. An advantage over the result of Abbasi-Yadkori et al. (2011) is that the term $\sigma_{\text{var}}^2$ is always smaller than $\sigma_{\text{subG}}^2$ appearing in (2). Moreover, even though both bounds assume oracle access to the variance (proxy), a second order statistic such as $\sigma_{\text{var}}^2$ is more amenable to be directly estimated from data.

2. When $\Gamma = 0$, requiring $\alpha = 0$ in Theorem 1 can be thought of as a burn-in requirement, restricting the Bernstein inequality to cases in which the corresponding least squares error is sufficiently small. Typically, $\alpha = 0$ once the sample size is large enough as $\|S_\tau\|_{V_\tau^{-2}}^2$ is the norm-squared error of the least squares estimator in the model $Y_k = \langle \theta_\star, X_k \rangle + W_k$ up to time $\tau$. When $S_\tau$ is not too large, the variance proxy $\sigma_{\text{subG}}^2$ in (2) can thus be replaced by the variance term $\sigma_{\text{var}}^2$ with just a little overhead in $\varepsilon$.

3. For $\Gamma \succeq \varepsilon^{-1}e^{-1}(d+2)B_W^2 B_X^2$ (corresponding instead to ridge regression) one may choose $V = \Gamma$ and use (2) to control $\alpha$ at the cost of an inflated failure probability ($\delta$ to $2\delta$).

4. Together, $\varepsilon$ and $\nu$ control the sharpness of the multiplicative constants of the bound. In the large sample regime, when $\|S_\tau\|^2_{V_\tau^{-2}}$ is small, these can often both be allowed to tend to 0.

5. The bound can be put in a more convenient form by making use of either of the two numerical inequalities $(1 + 2\alpha)^{-1}(1 + \alpha)^2 \le (1 + \alpha^2) \wedge (1 + \frac{1}{2}\alpha)$.

6. Using Theorem 1 instead of the result in Abbasi-Yadkori et al. (2011) sharpens existing bounds for least squares estimators with martingale difference noise, allowing one to achieve optimal dependence in $\sigma^2_{\mathrm{var}}$. See e.g. the proof structure laid out in Ziemann et al. (2023, Section 1.2).

7. These improvements also potentially extend to refining linear bandit analyses, previously motivating the results in Abbasi-Yadkori et al. (2011) and Zhao et al. (2023).

## 3 PAC-Bayesian Bounds

Let $\rho$ and $\pi$ be two probability measures supported on a set $\Lambda \subset \mathbb{R}^d$. Recall that $\mathrm{d}_{\mathrm{KL}}(\rho, \pi) \triangleq \int_\Lambda \log\left(\frac{d\rho}{d\pi}\right) d\rho \in [0, +\infty]$ is the Kullback-Leibler divergence between $\rho$ and $\pi$. Our analysis in the sequel rests on the well-known PAC-Bayesian lemma, stated below.

**Lemma 1** (PAC-Bayesian deviation bound). *Let $\Lambda$ be a subset of $\mathbb{R}^d$, and $Z(\lambda)$, $\lambda \in \Lambda$, be a family of real-valued random variables. Assume that $\mathbf{E}[\exp Z(\lambda)] \le 1$ for every $\lambda \in \Lambda$. Let $\pi$ be a probability distribution on $\Lambda$. Then for all $u \in [0, \infty)$ :*

$$\mathbf{P}\left(\forall \rho : \int_\Lambda Z(\lambda)\mathrm{d}\rho(\lambda) \le \mathrm{d}_{\mathrm{KL}}(\rho, \pi) + u\right) \ge 1 - e^{-u}, \tag{5}$$

*where $\rho$ spans all probability measures on $\Lambda$.*

We will instantiate the PAC-Bayesian lemma with $Z(\lambda)$ as the quadratic form (and with $t = \tau$)

$$Z(\lambda) = \langle \lambda, S_t \rangle - \frac{1}{2}\|\lambda\|^2_{V_t} = \frac{1}{2}\|S_t\|^2_{V_t^{-1}} - \frac{1}{2}\|\lambda - V_t^{-1}S_t\|^2_{V_t}. \tag{6}$$

To account for the fact that the Mahalanobis norm appearing in (2) includes an additive factor $\Gamma$, let us also take note of the following identity, which can be obtained by completing the square in (6):

$$\|S_t\|^2_{V_t^{-1}} - \|\lambda - V_t^{-1}S_t\|^2_{V_t} = \|S_t\|^2_{(V_t+\Gamma)^{-1}} - \|\lambda - (V_t + \Gamma)^{-1}S_t\|^2_{V_t+\Gamma} + \|\lambda\|^2_\Gamma. \tag{7}$$

In particular, we seek to bound the RHS of (6) (or (7)) but will use the LHS to establish an exponential inequality. In the two sections that follow we first show how to recover the results of Abbasi-Yadkori et al. (2011) featuring the variance proxy $\sigma^2_{\mathrm{subG}}$ and then proceed to generalize these to variance sensitive Bernstein bounds depending on $\sigma^2_{\mathrm{var}}$ in the leading order.

### 3.1 Warm-up: Sub-Gaussian Deviation Bounds

Before we prove our main result, let us explain how a version of the result of Abbasi-Yadkori et al. (2011) can be established via Lemma 1. This will inform the proof strategy of our result by essentially replacing Gaussians with a certain covariance ellipsoid with uniform distributions over the same ellipsoid. We prove the result for $\sigma^2_{\mathrm{subG}} = 1$ and note that the general case follows by rescaling.

By making use of the identity (6), it is easy to see that the right hand side of (7) satisfies the exponential inequality required for Lemma 1. Namely, the tower rule and the conditional sub-Gaussianity of $\{W_k, k \ge 1\}$ implies that $\mathbf{E}\exp\left(\langle \lambda, S_T \rangle - \frac{1}{2}\|\lambda\|^2_{V_T}\right) \le 1$ for all $\lambda \in \mathbb{R}^d$ and $T \in \mathbb{N}$. Hence, we may pick

$\rho = \mathsf{N}\left((V_T + \Gamma)^{-1} S_T, \Sigma_\rho\right)$ in Lemma 1. We have:

$$\frac{1}{2} \int \left[ \|S_T\|_{(V_T+\Gamma)^{-1}}^2 - \|\lambda - (V_T + \Gamma)^{-1} S_T\|_{V_T+\Gamma}^2 + \|\lambda\|_\Gamma^2 \right] \mathrm{d}\rho(\lambda)$$

$$= \frac{1}{2} \left[ \|S_T\|_{(V_T+\Gamma)^{-1}}^2 - \mathrm{tr}((V_T + \Gamma)\Sigma_\rho) + \mathrm{tr}\left(\Gamma\Sigma_\rho + \Gamma(V_T + \Gamma)^{-1} S_T S_T^{\mathsf{T}} (V_T + \Gamma)^{-1}\right) \right]$$

$$= \frac{1}{2} \left[ \|S_T\|_{(V_T+\Gamma)^{-1}}^2 - \mathrm{tr}(V_T \Sigma_\rho) + \|S_T\|_{(V_T+\Gamma)^{-1}\Gamma(V_T+\Gamma)^{-1}}^2 \right] \quad (8)$$

Moreover, if we now set $\pi = \mathsf{N}(0, \Sigma_\pi)$ we have that:

$$\mathrm{d}_{\mathrm{KL}}(\rho, \pi) = \frac{1}{2} \left[ \mathrm{tr}(\Sigma_\pi^{-1}\Sigma_\rho - I) + \|S_T\|_{(V_T+\Gamma)^{-1}\Sigma_\pi^{-1}(V_T+\Gamma)^{-1}}^2 + \log \frac{\det \Sigma_\pi}{\det \Sigma_\rho} \right] \quad (9)$$

We point out that there is an asymmetry between $\Sigma_\rho$ and $\Sigma_\pi$ at this point. The first is allowed to depend on the processes $S_T, V_T$ but the latter is not. Applying Lemma 1 to the process (6) yields that with probability at least $1 - e^{-u}$:

$$\frac{1}{2} \left[ \|S_T\|_{(V_T+\Gamma)^{-1}}^2 - \mathrm{tr}(V_T \Sigma_\rho) + \|S_T\|_{(V_T+\Gamma)^{-1}\Gamma(V_T+\Gamma)^{-1}}^2 \right]$$

$$\leq \frac{1}{2} \left[ \mathrm{tr}(\Sigma_\pi^{-1}\Sigma_\rho - I) + \|S_T\|_{(V_T+\Gamma)^{-1}\Sigma_\pi^{-1}(V_T+\Gamma)^{-1}}^2 + \log \frac{\det \Sigma_\pi}{\det \Sigma_\rho} \right] + u. \quad (10)$$

Equivalently:

$$\|S_T\|_{(V_T+\Gamma)^{-1}}^2 + \|S_T\|_{(V_T+\Gamma)^{-1}\Gamma(V_T+\Gamma)^{-1}}^2 - \|S_T\|_{(V_T+\Gamma)^{-1}\Sigma_\pi^{-1}(V_T+\Gamma)^{-1}}^2$$

$$\leq \mathrm{tr}(\Sigma_\pi^{-1}\Sigma_\rho + V_T \Sigma_\rho - I) + \log \frac{\det \Sigma_\pi}{\det \Sigma_\rho} + 2u. \quad (11)$$

Hence it makes sense to choose $\Sigma_\rho = (V_T + \Gamma)^{-1}$ and $\Sigma_\pi = \Gamma^{-1}$ giving:

$$\|S_T\|_{(V_T+\Gamma)^{-1}}^2 \leq \log \frac{\det(V_T + \Gamma)}{\det(\Gamma)} + 2u \quad (12)$$

which is identical to the result of Abbasi-Yadkori et al. (2011) (modulo the stopping time, which can easily be addressed—see below).

**Remark 3.1.** *By directly applying this proof strategy to* (6) *one may also obtain the following deviation bound with probability at least $1 - e^{-u}$:*

$$\|S_T\|_{V_T^{-1}}^2 - \|S_T\|_{V_T^{-1}\Gamma V_T^{-1}}^2 \leq \log \frac{\det(V_T)}{\det(\Gamma)} + 2u. \quad (13)$$

### 3.2 Variance Sensitive Deviation Bounds: Proof of Theorem 1

The use of Gaussian distributions in the self-normalized martingale bound is very convenient as it admits closed form KL expressions. However, their use hinges on the fact that an exponential inequality holds throughout $\mathbb{R}^d$. We now show how to obtain a similar bound using distributions with compact support. In the sequel, we assume that $\sigma_{\mathrm{var},\varepsilon}^2 \triangleq (1-\varepsilon)^{-1}\sigma_{\mathrm{var}}^2 = 1$ and note that the general result can be recovered by rescaling $S_\tau$.

Let us now consider two ellipsoidal distributions. We construct them as follows. Fix two positive definite matrices $\Sigma_\pi$ and $\Sigma_\rho$ to be determined momentarily and a measurable weight factor $\alpha \in [0, \infty)$. First, let $\pi$ be uniform over the ellipsoid centered at zero and with shape $\Sigma_\pi$, i.e., uniform over $\{x \in \mathbb{R}^d : x^{\mathsf{T}}\Sigma_\pi^{-1}x \leq 1\}$. Second, let $\rho$ be uniform over the ellipsoid with center $\frac{1}{1+\alpha}(V_\tau + \Gamma)^{-1}S_\tau$ and shape $\Sigma_\rho$. Note that we must choose $\alpha$ such that $\frac{1}{1+\alpha}(V_\tau + \Gamma)^{-1}S_\tau + \{x \in \mathbb{R}^d : x^{\mathsf{T}}\Sigma_\rho^{-1}x \leq 1\} \subset \{x \in \mathbb{R}^d : x^{\mathsf{T}}\Sigma_\pi^{-1}x \leq 1\}$. Momentarily

leaving this point aside, it is easy to see that on the event that the second ellipsoid is contained in the first, the KL divergence between these two distributions is the logarithmic volume ratio (Lemma 4):

$$\rho - \text{Ellipsoid} \subset \pi - \text{Ellipsoid} \quad \Longrightarrow \quad d_{\text{KL}}(\rho, \pi) = \frac{1}{2} \log \frac{\det \Sigma_\pi}{\det \Sigma_\rho}. \tag{14}$$

Moreover, using Lemma 3, on the same event we have that

$$\frac{1}{2} \int \left[ \|S_\tau\|^2_{(V_\tau+\Gamma)^{-1}} - \|\lambda - (V_\tau+\Gamma)^{-1} S_\tau\|^2_{V_\tau+\Gamma} + \|\lambda\|^2_\Gamma \right] d\rho(\lambda)$$

$$= \frac{1}{2} \left[ \frac{1+2\alpha}{(1+\alpha)^2} \|S_\tau\|^2_{(V_\tau+\Gamma)^{-1}} - \frac{1}{d+2} \text{tr}(V_\tau \Sigma_\rho) + \frac{1}{(1+\alpha)^2} \|S_\tau\|^2_{(V_\tau+\Gamma)^{-1}\Gamma(V_\tau+\Gamma)^{-1}} \right] \tag{15}$$

In other words, combined with $\|S_\tau\|_{(V_\tau+\Gamma)^{-1}\Gamma(V_\tau+\Gamma)^{-1}} \geq 0$, the PAC-Bayesian bound in Lemma 1, justified by the exponential inequality in Lemma 2, yields that with probability $1 - e^{-u}$:

$$\frac{1+2\alpha}{(1+\alpha)^2} \|S_\tau\|^2_{(V_\tau+\Gamma)^{-1}} \leq 2u + \frac{1}{d+2} \text{tr}(V_\tau \Sigma_\rho) + \log \frac{\det \Sigma_\pi}{\det \Sigma_\rho}. \tag{16}$$

In particular, we may choose $\Sigma_\rho = (d+2)(V_\tau+\Gamma)^{-1}$ and $\Sigma_\pi = e^{-1}(d+2)V^{-1}$ to obtain:

$$\frac{1+2\alpha}{(1+\alpha)^2} \|S_\tau\|^2_{(V_\tau+\Gamma)^{-1}} \leq 2u + \log \frac{\det(V_\tau+\Gamma)}{\det(V)}. \tag{17}$$

We note that (16)-(17) only hold when the event $\{\rho - \text{Ellipsoid} \subset \pi - \text{Ellipsoid}\}$ occurs, which in itself is contingent on our choice of $\alpha$. First, note that with our choice of priors, the use of Lemma 2 requires the additional constraint $V \succeq \varepsilon^{-1}e^{-1}(d+2)B_W^2 B_X^2$. To conclude the proof it remains to verify that a good choice of $\alpha$ can be made. The required event occurs precisely when

$$\frac{1}{1+\alpha}(V_\tau+\Gamma)^{-1} S_\tau + \{x \in \mathbb{R}^d : x^\mathsf{T}(V_\tau+\Gamma)x \leq d+2\} \subset \{x \in \mathbb{R}^d : x^\mathsf{T} V x \leq e^{-1}(d+2)\}. \tag{18}$$

If $V_\tau + \Gamma \succeq (1+\nu)^2 e V$ it suffices

$$\left( \frac{\sqrt{d+2}}{1+\nu} + \frac{1}{1+\alpha} \|S_\tau\|_{(V_\tau+\Gamma)^{-1}V(V_\tau+\Gamma)^{-1}} \right)^2 \leq e^{-1}(d+2). \tag{19}$$

To see this, let $y = \frac{1}{1+\alpha}(V_\tau+\Gamma)^{-1} S_\tau$ and note that we must show that $(y+x)^\mathsf{T} V(y+x) \leq e^{-1}(d+2)$ for every $x \in \mathbb{R}^d$ satisfying $x^\mathsf{T}(V_\tau+\Gamma)x \leq d+2$. Now we have that for every $\mu > 0$, using Young's inequality and optimizing the weight $\mu$ below:

$$\begin{aligned}
(y+x)^\mathsf{T} V(y+x) &\leq (1+\mu)x^\mathsf{T} V x + (1+\mu^{-1})y^\mathsf{T} V y \\
&\leq \frac{(1+\mu)(d+2)}{e(1+\nu)^2} + (1+\mu^{-1})y^\mathsf{T} V y \qquad (V \preceq e^{-1}(1+\nu)^{-2}(V_\tau+\Gamma)) \\
&= \left( \frac{\sqrt{d+2}}{\sqrt{e}(1+\nu)} + \sqrt{y^\mathsf{T} V y} \right)^2 \qquad \left( \mu = \sqrt{\frac{e(1+\nu)^2 y^\mathsf{T} V y}{d+2}} \right).
\end{aligned} \tag{20}$$

Moreover, one can verify that the above inequality holds with $\alpha = \left( \frac{\sqrt{e}(1+\nu)\|S_\tau\|_{(V_\tau+\Gamma)^{-1}V(V_\tau+\Gamma)^{-1}}}{\nu\sqrt{d+2}} - 1 \right) \vee 0$, so that our priors are indeed well defined for this choice. Hence, under the imposed constraints on $V, \varepsilon$ and $\nu$ we have thus obtained that

$$\frac{1+2\alpha}{(1+\alpha)^2} \|S_\tau\|^2_{(V_\tau+\Gamma)^{-1}} \leq 2u + \log \frac{\det(V_\tau+\Gamma)}{\det(V)}. \tag{21}$$

This finishes the proof. ∎

## 4 Auxiliary Results

**Lemma 2.** *Impose* (3), *fix a stopping time $\tau$ with respect to $\mathcal{F}_{0:\infty}$ and $\varepsilon \in (0,1)$. We have that*

$$\mathbf{E} \exp \left( \langle \lambda, S_\tau \rangle - \frac{\sigma_{\mathrm{var},\varepsilon}^2 \|\lambda\|_{V_\tau}^2}{2} \right) \leq 1 \tag{22}$$

*for every $\lambda \in \mathbb{R}^d$ satisfying $\|\lambda\|_{B_X^2 B_W^2}^2 \leq \varepsilon^2$ and where $\sigma_{\mathrm{var},\varepsilon}^2 \triangleq (1-\varepsilon)^{-1} \sigma_{\mathrm{var}}^2$.*

*Proof.* Applying Bernstein's moment inequality conditionally on $\mathcal{F}_{k-1}$ yields that

$$\mathbf{E}_{k-1}[e^{\langle \lambda, k_t \rangle W_k}] \leq \exp \left( \frac{\|\lambda\|_{X_k X_k^\mathsf{T}}^2 \sigma_{\mathrm{var}}^2}{2(1 - |\langle \lambda, X_k \rangle| B_W)} \right) \tag{23}$$

as long as $\lambda^\mathsf{T} X_k X_k^\mathsf{T} \lambda = |\langle \lambda, X_k \rangle|^2 < B_W^{-2}$. Since $X_k X_k^\mathsf{T} \preceq B_X^2$, this holds deterministically as long as $\|\lambda\|_{B_W^2 B_X^2}^2 < 1$. In particular, if we fix $\varepsilon \in (0,1)$ and impose $\|\lambda\|_{B_X^2 B_W^2}^2 \leq \varepsilon^2$ we find that

$$\mathbf{E}_{k-1}[e^{\langle \lambda, X_k \rangle W_k}] \leq \exp \left( \frac{\|\lambda\|_{X_k X_k^\mathsf{T}}^2 \sigma_{\mathrm{var}}^2}{2(1-\varepsilon)} \right) \tag{24}$$

Applying the tower property repeatedly, it thus follows that with $\sigma_{\mathrm{var},\varepsilon}^2 = (1-\varepsilon)^{-1} \sigma_{\mathrm{var}}^2$ we have that

$$\mathbf{E} \exp \left( \langle \lambda, S_t \rangle - \frac{\sigma_{\mathrm{var},\varepsilon}^2 \|\lambda\|_{V_t}^2}{2} \right) \leq 1 \tag{25}$$

for every $t$ and for every $\lambda$ satisfying $\|\lambda\|_{B_X^2 B_W^2}^2 \leq \varepsilon^2$.

To prove the result for $\tau$ a stopping time, define the (by the calculations above) nonnegative supermartingale $M_t \triangleq \exp \left( \langle \lambda, S_t \rangle - \frac{\sigma_{\mathrm{var},\varepsilon}^2 \|\lambda\|_{V_t}^2}{2} \right)$. It follows by standard optional stopping arguments and Fatou's Lemma that $M_\tau = \liminf_{T \to \infty} M_{\tau \wedge T}$ also is a nonnegative supermartingale. In particular $\mathbf{E} M_\tau \leq 1$ as per requirement. ∎

**Lemma 3.** *Fix $\Sigma \in \mathbb{R}^{d \times d}, \Sigma \succ 0$ and let $U$ be uniformly distributed over $\{x \in \mathbb{R}^d | x^\mathsf{T} \Sigma^{-1} x \leq 1\}$. Then $\mathbf{E} U U^\mathsf{T} = \frac{1}{d+2} \Sigma$.*

*Proof.* Since $U = \sqrt{\Sigma} Y$ where $Y$ is uniform over $x^\mathsf{T} x \leq 1$ it suffices to prove the result for $Y$. By symmetry we must have $\mathbf{E} Y Y^\mathsf{T} = \alpha I_d$ for some $\alpha > 0$. Moreover, it is easy to see that $\mathrm{tr}\, \mathbf{E} Y Y^\mathsf{T} = \frac{d}{d+2}$. Hence $\alpha = (d+2)^{-1}$ and the result is established. ∎

**Lemma 4.** *Fix $\Sigma_\pi, \Sigma_\rho \in \mathbb{R}^{d \times d}$ with $\Sigma_\pi, \Sigma_\rho \succ 0$. Let $\pi$ and $\rho$ be uniform distributions over $E_\pi \triangleq \{x \in \mathbb{R}^d | x^\mathsf{T} \Sigma_\pi^{-1} x \leq 1\}$ and $E_\rho \triangleq \{x \in \mathbb{R}^d | x^\mathsf{T} \Sigma_\rho^{-1} x \leq 1\}$ respectively. If $E_\rho \subseteq E_\pi$ we have that $\mathrm{d}_{\mathrm{KL}}(\rho, \pi) = \frac{1}{2} \log \frac{\det \Sigma_\pi}{\det \Sigma_\rho}$.*

*Proof.* We have that

$$\begin{aligned}
\mathrm{d}_{\mathrm{KL}}(\rho, \pi) &= \int_{E_\rho} \log \frac{\rho(x)}{\pi(x)} \mathrm{d}\rho(x) \\
&= \int_{E_\rho} \log \frac{\det \sqrt{\Sigma_\pi}}{\det \sqrt{\Sigma_\rho}} \mathrm{d}\rho(x) && \text{(volume ratio)} \\
&= \log \frac{\det \sqrt{\Sigma_\pi}}{\det \sqrt{\Sigma_\rho}} \int_{E_\rho} \mathrm{d}\rho(x) = \log \frac{\det \sqrt{\Sigma_\pi}}{\det \sqrt{\Sigma_\rho}} && \left( \int_{E_\rho} \mathrm{d}\rho(x) = 1 \right)
\end{aligned} \tag{26}$$

as per requirement. ∎

*Proof of Lemma 1.* By integrating the inequality $\mathbf{E}[\exp Z(\lambda)] \leq 1$ with respect to $\pi$ and Fubini:

$$\mathbf{E}\left[\int_\Lambda \exp Z(\lambda)\mathrm{d}\pi(\lambda)\right] \leq 1. \tag{27}$$

We now change measure using the variational characterization of the relative relative entropy functional (Donsker & Varadhan, 1975), which reads:

$$\log \int_\Lambda \exp(Z(\lambda))\mathrm{d}\pi(\lambda) = \sup_\rho \left\{\int_\Lambda Z(\lambda)\mathrm{d}\rho(\lambda) - \mathrm{d}_{\mathrm{KL}}(\rho, \pi)\right\}, \tag{28}$$

where the supremum spans over all probability measures $\rho$ over $\Lambda$. Hence

$$\mathbf{E}\left[\exp \sup_\rho \left\{\int_\Lambda Z(\lambda)\mathrm{d}\rho(\lambda) - \mathrm{d}_{\mathrm{KL}}(\rho, \pi)\right\}\right] \leq 1. \tag{29}$$

The result follows by a Chernoff bound applied to $\left\{\sup_\rho \left\{\int_\Lambda Z(\lambda)\mathrm{d}\rho(\lambda) - \mathrm{d}_{\mathrm{KL}}(\rho, \pi)\right\} > u\right\}$. ∎

## Acknowledgments

The author acknowledges support by a Swedish Research Council international postdoc grant and thanks Bruce Lee for several stimulating discussions and Alberto Metelli for pointing out a miscalculation in an earlier version of the manuscript.

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
