# OpenReview forum: "A Vector Bernstein Inequality for Self-Normalized Martingales"
_TMLR — Accepted by TMLR_

### Review · Reviewer_gDu1 · 2025-02-09

**Summary Of Contributions:**

The authors put forward a new technique to prove deviations inequalities for so-called self-normalized martingales.
The novelty is that the proposed technique relies on the PAC-Bayesian lemma.

1/ The authors first demonstrate that their technique is able to recover a state-of-the-art bound from Abbasi-Yadkori (2011).
The above-mentioned bounds relies on an assumption: a bound on the sub-Gaussian norm of the process (conditioned on the past).

2/ The authors then propose an improvement over the previous known result by obtaining an alternative bound where a
bound on the sub-Gaussian norm is replaced with the conditional variance.

**Audience:**

Yes

**Claims And Evidence:**

Yes

**Requested Changes:**

The changes I propose are only minor and are as follows:

1/ Since the paper is quite short, I would suggest that the authors take this opportunity to add a few more intermediary steps in their proofs (e.g. how to obtain the identity at (7)).

2/ Perhaps review a bit the structure of the paper.
    There currently is a single sub-section in Section 1.

3/ Add a few more references:

a) I appreciate that the authors included a short derivation of the PAC-Bayesian lemma at the end of the work.
Perhaps they can include a reference to most well-cited usages of this lemma in the literature.
Also mention after stating the lemma that you will offer a short proof at the end of the script.

b) On page 1: "The traditional fix in the literature", please add a few references where this analysis is carried out.

4/ On p4., Section 2.2, forth line. Was  $\sigma^2_{var,\varepsilon}$ defined anywhere before equation (22)?

5/ In Remarks on Theorem 1, 3. When $\nu \to 0$, don't you have that $\alpha \to \infty$, and so does your bound? What do I miss?

**Strengths And Weaknesses:**

* Strengths:

1/ The content is relatively self-contained. I like that the authors present their strategy.

2/ The result is concise / crisp.

3/ The argument of the authors that their bound will find applications is plausible.
    The authors chose to not directly illustrate this claim directly (e.g. for bandit problems), but I think it is not necessary.

* Weaknesses:

1/ Could consider a few additional steps in the derivations to help the reader.

2/ I think the reader would also appreciate a few more references for some of the tools and claims being made.

---

> ### Author Response · Authors · 2025-02-15
>
> Many thanks for taking the time to review the manuscript and for your thoughtful comments!
>
> I have taken the following steps to address your suggested changes:
>
> * added a short note on how to obtain equation (7). If there are any further derivations you think need clarification I am happy to adress these as well.
>
> * Broke away subsection 1.1 into its own section (now sec 2).
>
> * added references: Freedman/De la Pena (bernstein-type bounds) & McAllester / Shawe-Taylor (PAC-Bayes)
>
> * defined the variance proxy in section 2.2
>
> Regarding your last point on the remark 3: I have clarified my use of "large sample regime" to mean that the corresponding least squares error is small. In this regime the numerator in alpha is also small. Essentially one should think of the remark as suggesting a "proportional limit".

---

> > ### Comment · Reviewer_gDu1 · 2025-02-17
> >
> > Thank you very much for addressing my comments and for the clarification regarding [Remarks on Theorem 1].

---

### Review · Reviewer_4FTh · 2025-02-09

**Summary Of Contributions:**

This paper establishes a new Bernstein inequality for vector-valued self-normalized martingales, providing a concentration bound that depends on the variance rather than a uniform bound. The proof is based on a PAC–Bayesian argument with Gaussian priors, which also yields a new proof of a sub-Gaussian bound by Abbasi-Yadkori et al. This approach appears to be novel.

**Audience:**

Yes

**Broader Impact Concerns:**

None.

**Claims And Evidence:**

Yes

**Requested Changes:**

Provide additional examples or applications to illustrate how the new Bernstein inequality can be applied in concrete settings.

Lemma 1 is not clearly stated; in the expression $\mathbf E[\exp Z(\lambda)] \le 1$ it should be specified that $\lambda$ is drawn from $\pi$.

In Theorem 1, "we that" is a typo.

**Strengths And Weaknesses:**

Strengths:
- The main result is a nice theoretical advance for the study of concentration inequalities which are broadly applied.
- The paper is generally well-written and well-structured.

Weaknesses:
- The paper would benefit from additional examples or concrete applications to illustrate the impact of the new bound.

---

> ### Author Response · Authors · 2025-02-15
>
> Many thanks for taking the time to review the manuscript and for your thoughtful comments!
>
> * I have added an additional mention of an application/interpretation of the result as per your request. The result actually immediately implies a confidence ellipsoid for least squares estimation, and as such I don't think further downstream applications are necessary. I have however clarified this by adding an addition remark immediately following Theorem 1.
>
> * Regarding Lemma 1 there appears to be some misunderstanding on account of the reviewer. As it stated in the theorem, $\lambda$ is fixed in the hypothesis (the mgf bound must hold pointwise for every such entity). Integration (expectation) is thus carried out only over the randomness over $Z(\lambda)$ which is a fixed random variable.

---

> > ### Comment · Reviewer_4FTh · 2025-02-15
> > **Response**
> >
> > Thank you for the clarification, indeed it was my misunderstanding.

---

### Review · Reviewer_5zFA · 2025-02-17

**Summary Of Contributions:**

This is a technical paper where the authors prove a Bernstein type inequality for vector-valued random variables.

**Audience:**

No

**Broader Impact Concerns:**

See my previous comment. I am strongly recommending a clear cut rejection.

**Claims And Evidence:**

No

**Requested Changes:**

I suggest to send the paper to an applied probability journal. However, the mathematical level is definitely not up to the top ones - I would recommend the Journal of Applied Probability as a good venue for this paper.

**Strengths And Weaknesses:**

The authors are mathematically competent and the mathematical narrative is neat.

On the other hand, this is a probability paper and definitely it does not fit TMLR. The motivations are definitely weak, and I foresee no applications of the findings exposed here, except for a loose comment on ridge regression.

---

> ### Author Response · Authors · 2025-02-17
>
> Many thanks for taking the time to review the manuscript.
>
> It appears to me that the reviewer's main criticism is the scope of the paper. I disagree for the following reasons:
>
> * First, while TMLR is relatively newer, machine learning conferences have a history of publishing this type of paper (concentration inequality with application to learning algorithm). Some examples are [A,B] below. Moreover, from my reading of the submission guidelines of TMLR this appears aligned with the scope of the journal.
>
> * Second, The result is as it stands already a confidence bound for the least squares estimator applied to data with martingale difference noise---there only being loose connections to applications is not true. It is directly applicable as it stands and I have clarified this point in the updated version of the manuscript posted earlier this weekend.
>
> * In a little more detail, let me reiterate that this type of bound (conc. ineq. for vector-valued self-normalized martingales) is often a key lemma in two lines of work: analysis of reinforcement learning algorithms (linear bandits in particular) and the analysis of learning algorithms coming from sequentially dependent data streams. The present bound is a tight Bernstein inequality where there previously only was a tight Hoeffding inequality.
>
>    * I am not an expert on linear bandits, but it appears to me that concentration inequalities for self-normalized martingales have been a key component in their analysis for a long time. This is also stated in the manuscipt and appears to have been missed by the reviewer. Indeed the first theorem (~main technical lemma) in paper [C], which is one of the most cited references on linear bandits (over 2000 citations), is precisely the Hoeffding bound that we generalize to a variance sensitive bound. There have been a number of follow-up works to [C] in which relatively looser Bernstein-type inequalities feature as a key technical lemma (see e.g. [D, E]).
>
>   * The bound also constitutes roughly (1/2) of the analysis necessary for proving guarantees on realizable least squares with dependent data. In the model $Y_t = \theta_\star X_t +W_t$, ($\theta_\star$ a matrix) one can express the least squares error as: $$  \widehat \theta  -\theta^\star = \left[\left(\sum_{t=1}^{T} V_t X_t^\top \right)\left(\sum_{t=1}^{T} X_t X_t^\top \right)^{-1/2 }\right]\left(\sum_{t=1}^{T} X_t X_t^\top \right)^{-1/2 }.$$ Our bound controls the term in square brackets and the other half of the analysis consists of the controlling the lower tail of the random design $\left(\sum_{t=1}^{T} X_t X_t^\top \right)^{-1/2 }$. As I pointed out in the manuscript, applying the presented bound here yields an improvement from a sub-Gaussian error-dependency to a variance-sensitive error-dependency.  Moreover, this type of quotient also appears in the analysis of more general nonlinear model classes by a typical M-estimation argument (taylor expansion).
>
>
> [A] Kontorovich, Aryeh. "Concentration in unbounded metric spaces and algorithmic stability." International conference on machine learning. PMLR, 2014.
>
> [B] Maurer, Andreas, and Massimiliano Pontil. "Concentration inequalities under sub-gaussian and sub-exponential conditions." Advances in Neural Information Processing Systems 34 (2021): 7588-7597.
>
> [C] Abbasi-Yadkori, Yasin, Dávid Pál, and Csaba Szepesvári. "Improved algorithms for linear stochastic bandits." Advances in neural information processing systems 24 (2011).
>
> [D] Zhou, Dongruo, Quanquan Gu, and Csaba Szepesvari. "Nearly minimax optimal reinforcement learning for linear mixture markov decision processes." Conference on Learning Theory. PMLR, 2021.
>
> [E] Zhao, Heyang, et al. "Variance-dependent regret bounds for linear bandits and reinforcement learning: Adaptivity and computational efficiency." The Thirty Sixth Annual Conference on Learning Theory. PMLR, 2023.

---

### Decision · Action_Editor_zQH8 · 2025-03-06

**Recommendation:** Accept as is

**Comment:**

The TMLR acceptance criteria are that:

- (Claims and evidence) All claims in the paper are supported by accurate and convincing evidence;
- (Audience) And that *some* part of the community would be interested in the results.

Reviewers all agree that the result is rigorously proven, and they enjoyed reading through the discussion of the proof strategy prior to seeing the calculations.
Thus the paper clearly satisfies (Claims and evidence).

As for audience, while one of the reviewers has some concerns about the significance, impact and scope of the result, the other two reviewers think that the current results are already of interest to *some* people. I also agree that a new concentration bound, with new proof technique, is sufficient to pass the (Audience) criterion.

I recommend acceptance of this paper to TMLR.

**Audience:**

Yes, concentration inequalities are fundamental tools that are of interest to the theoretical parts of the community.

**Claims And Evidence:**

Reviewers unanimously think that the paper is well-structured and gives a clear proof of the claimed concentration bound.

---

> ### Author Response · Authors · 2025-03-21
>
> Thanks for a very pleasant submission experience!
>
> Please find the camera ready uploaded through OpenReview.